

# Soil properties and root traits jointly shape fine-scale spatial patterns of bacterial community and metabolic functions within a Korean pine forest

Jialing Teng[1,2], Jing Tian[3], Guirui Yu[1,2] and Yakov Kuzyakov[4,5]

[1] Key Laboratory of Ecosystem Network Observation and Modeling, Institute of Geographic Sciences and Natural Resources Research, Chinese Academy of Sciences, Beijing, China
[2] College of Resources and Environment, University of Chinese Academy of Sciences, Beijing, China
[3] College of Resources and Environmental Sciences; Key Laboratory of Plant-Soil Interactions, Ministry of Education, China Agricultural University, Beijing, China
[4] Department of Soil Science of Temperate Ecosystems, University of Göttingen, Göttingen, Germany
[5] Institute of Environmental Sciences, Kazan Federal University, Kazan, Russia

Corresponding authors
Jing Tian, tianj@cau.edu.cn
Guirui Yu, yugr@igsnrr.ac.cn

## ABSTRACT

Spatial heterogeneity of soil bacterial community depends on scales. The fine-scale spatial heterogeneity of bacterial community composition and functions remains unknown. We analyzed the main driving factors of fine-scale spatial patterns of soil bacterial community composition and carbon metabolic functions across a 30 m × 40 m plot within a Korean pine forest by combining Illumina 16S rRNA sequencing with Biolog Ecoplates based on 53 soil samples. Clear spatial patterns in bacterial community composition and metabolic functions were observed in the forest soil. The bacterial community composition and metabolic functions both showed distance-decay of similarity within a distance of meters. Structural equation model analysis revealed that environmental variables and geographic distance together explained 37.9% and 63.1% of community and metabolic functions, respectively. Among all environmental factors, soil organic carbon (SOC) and root biomass emerged as the most important drivers of the bacterial community structure. In contrast, soil pH explained the largest variance in metabolic functions. Root biomass explained the second-largest variance in soil bacterial community composition, but root traits made no difference in metabolic functions variance. These results allow us to better understand the mechanisms controlling belowground diversity and plant-microbe interactions in forest ecosystems.

## INTRODUCTION

Soil bacteria drive important biogeochemical processes and play critical roles in regulating the functions and stability of ecosystems (*Fuhrman, 2009*; *Lladó, López-Mondéjar & Baldrian, 2017*; *Sugden, 2018*). The geographic distribution of soil bacteria has been recently examined across a broad range of spatial scales (*Griffiths et al., 2011*; *Martiny et al., 2011*; *Ranjard et al., 2013*; *Sugden, 2018*). Most of these studies compared samples collected more than 1 km apart, and studies on fine-scale (<1 km) are still very rare (*Finkel et al.,*

*2012*; *Lear et al., 2014*). The fine-scale spatial patterns of soil bacteria are important for understanding bacterial community dynamics and providing appropriate scales to monitor the communities of forest soil. However, there are still important gaps in identifying the distances of patterns in community composition and diversity. In particular, the minimum spatial scales have significant biogeographic patterns (*Finkel et al., 2012*; *Lear et al., 2014*; *Martiny et al., 2006*).

Many environmental factors can directly or indirectly influence the spatial structure of soil bacterial communities. The soil pH has the clearest effects on the variance in the abundance of soil bacterial taxa (*Fierer, 2017*; *Liu et al., 2014*; *Shen et al., 2013*; *Tripathi et al., 2018*). The other most important factors influencing the structure of soil bacterial communities are quantity and quality of organic carbon (*Delgado-Baquerizo & Eldridge, 2019*; *Tian et al., 2012*; *Tian et al., 2018*), climate factors (*Bahram et al., 2018*; *Delgado-Baquerizo et al., 2017*; *Delgado-Baquerizo & Eldridge, 2019*; *Ladau et al., 2018*) and redox status (*DeAngelis et al., 2010*). Besides edaphic factors, plants are key drivers of soil bacterial community structure and functions (*Bardgett et al., 2005*; *DeVries et al., 2012*; *Delgado-Baquerizo et al., 2018a*; *Prober et al., 2015*). Plants have significant influences on carbon resources and modify the habitats of soil bacteria (*Kuzyakov, Friedel & Stahr, 2000*; *Latz et al., 2015*). Plant diversity is a strong predictor of soil microbial diversity (*Cantarel et al., 2016*; *Lamb et al., 2011*; *Prober et al., 2015*; *Wang et al., 2016*), and influence microbial communities via specific functional traits. Leaf traits (*Laughlin, 2011*) have important roles in regulating soil microbial communities (*De Vries et al., 2012*; *Delgado-Baquerizo et al., 2018b*). However, much less is known about the role of roots in regulating the soil bacterial communities (*De Vries et al., 2012*; *Delgado-Baquerizo et al., 2018b*; *Pervaiz et al., 2020*).

Forests are spatially heterogeneous ecosystems (*Stursova et al., 2016*) at scales of meters, within which soil, roots, and microbes form extremely complex dependencies and relationships through substance, energy, and information exchange (*Feeney et al., 2006*). The roots are one of the two dominant sources of C input variability in temperate forest soils (*Baldrian et al., 2010*; *Clemmensen et al., 2013*). Root-derived C forms easily available energy and supports a high abundance, activity, and diversity of microorganisms (*Pausch & Kuzyakov, 2011*). Active fine roots with fast turnover and their symbiotic microorganisms distribute throughout the soil and undertake essential functions for plants such as nutrient and water acquisition (*Finzi et al., 2015*; *Phillips & Fahey, 2006*; *Waisel et al., 2002*; *Zhao, Zeng & Fan, 2010*) and influence various ecological processes (*Bardgett & Wh, 2014*; *Cadotte et al., 2009*; *Clemmensen et al., 2013*; *Freschet & Cornelissen, 2013*). Active fine roots or tips can release more exudates into soil (*Dennis, Miller & Hirsch, 2010*; *Jones, Nguyen & Finlay, 2009*), which have an important influence on soil microbial communities (*Denef et al., 2009*; *Tian et al., 2012*).

Functional trait approaches have been demonstrated to be a beneficial tool for analyzing plant-microbial interactions (*Cantarel et al., 2016*; *De Vries et al., 2012*; *Grigulis et al., 2013*). For example, root diameter represents the ability of root to penetrate dense soil (*Materechera, Dexter & Alston, 1991*), colonizing by mycorrhiza (*Comas, Callahan & Midford, 2014*), whereas specific root length (SRL) reflects the efficiency of exploration or exploitation at the cost of root longevity (*Eissenstat et al., 2000*; *McCormack et al.,*

*2012*). Despite growing evidences that the effects of root traits on ecosystem processes largely via interactions with free-living microorganisms (*Bardgett, Mommer & De Vries, 2014*; *Freschet et al., 2017*), our knowledge of the specific traits that affect soil bacterial community composition and metabolic functions is limited.

In this context, we hypothesize that (1) bacterial community composition and carbon metabolic functions show distance-decay of similarity at a scale of few meters or tens of meters; and (2) environmental conditions, including soil properties and root traits explain more variance in bacterial community composition and metabolic functions than geographical distance because roots are the main direct drivers and the distance exerts indirect effects through the trees and their roots. To test these hypotheses, we investigated the significance of geographic distance, soil properties, and root traits in shaping the bacterial community composition and functions within a broad-leaved Korean pine forest.

## MATERIALS & METHODS

### Study site and sampling

The study was designed in an original Korean pine forest within the Forest Ecosystem Open Research Station of the Changbai Mountains in northeast China (28°28′E, 42°24′N) at an altitude of 700–800 m above sea level. This area is a typical warm temperate zone continental monsoon climate, with a mean air temperature of 2.0 °C and mean annual precipitation of approximately 700 mm. This region is dominated by brown forest soil, which originated from volcanic ash, and is classified as a Haplic Andosol (*Zhang, Han & Yu, 2006*). The vegetation community of the sampling plot is a multi-story forest with different ages, averaging over 200 years. The upper strata mainly include *Pinus koraiensis*, *Tiliaamurensis*, *Acer mono*, *Acer barbinerve*, *Fraxinus mandshurica*, *Acer ktegmentosum*, *Ulmus japonica*, and *Quercus mongolica*. The dominant shrubs and herbs include *Corylusmandshurica*, *Deutzia amurensis*, *Brachybotrysparidiformis*, and *Phrymaleptostachya*.

A total of 53 soil samples were collected (Fig. S1) from 0 to 10 cm depth from a 30 m × 40 m plot following the Latin hypercube design in August 2013, as described in *Tian et al. (2015)*. Latin hypercube sampling is a stratified-random procedure that provides an efficient way to ensure full coverage of the range of each variable by maximally stratifying the marginal distribution (*McKay, Beckman & Conover, 1979*). This design produces a statistically robust sampling scheme to capture the spatial variability of soils in the study area and is the most effective way to replicate the distribution of the variables (*Helton & Davis, 2003*; *Mulder, Bruin & Schaepman, 2013*). The sampling points had a minimum distance of 0.49 m and a maximum of 44 m. The samples were stored in airtight polypropylene bags, placed in a cooler box at about 4 °C during sampling and transported to the laboratory. The visible roots, rock fragments, and residues were carefully removed by hand, and then the roots were carefully washed with tap water to remove the adhering soil. Then samples were frozen at −20 °C until the measurements. The soil samples were divided into several subsamples. The subsamples for microbial functional diversity and dissolved organic matter concentration analysis were stored at 4 °C for no more than one

week. The subsamples for microbial communities were stored at −80 °C. The subsamples for organic matter analyses were air dried.

## Soil chemical analyses

The air-dried samples were passed through a two mm sieve, then ball-milled and analyzed for soil organic carbon (SOC) and total nitrogen (TN) contents by dry combustion with a Vario Max CN elemental analyzer (Elementar, Langenselbold, Germany). The soil dissolved organic carbon (DOC) and total dissolved nitrogen (DON) concentrations were determined using a Multi 3100 N/C TOC analyzer (Analytik Jena, Jena, Germany). The soil $NH_4^+$ and $NO_3^-$ concentrations were measured using an autoanalyzer (TRAACS-2000, BRAN+ LUEBBE, Norderstedt, Germany). The soil DON was calculated as the difference between the total dissolved N and the combined $NH_4^+$ and $NO_3^-$. The particulate organic carbon and nitrogen (POC and PON, respectively) were determined by the method reported by *Cambardella & Elliott (1992)*. Soil pH was determined using a pH meter after shaking the soil in deionized water (soil-to-water ratio of 1:2.5) suspensions for 30 min.

## Analyses of soil bacterial community composition and carbon metabolic functions

The soil microbial functional diversity was characterized using Biolog Eco-plates (Hayward, CA, USA) (*Garland & Mills, 1991*). Thirty-one C substrates associated with plant root exudates were used in the Eco-plates. Dividing them into six groups: seven carbohydrates ($\beta$-Methyl-D-glucoside, D-Xylose, i-Erythritol, d-Mannitol, N-Acetyl-$_D$-galactosamine, D-Cellobiose and $\alpha$-D-Lactose), six amino acids (L-Arginine, L-Asparagine, L-Phenylalanine, L-Serine, L-Threonine, and Glycyl- L-glutamic acid), nine carboxylic acids (D-Galactonic acid $\gamma$-lactone, D-Galacturonic acid, 2-Hydroxy benzoic acid, 4-Hydroxy benzoic acid, $\gamma$-Hydroxy butyric acid, Itaconic acid, $\alpha$-Keto butyric acid, D-Glucosaminic acid and D-Malic acid), two amines (Phenylethylamine and Putrescine), four polymers (Tween 40, Tween 80, $\alpha$-Cyclodextrin and Glycogen), and three miscellaneous (Pyruvic acid methy1 ester, D,L- $\alpha$-Glycerol phosphate and Glucose-L-phosphate). Briefly, 10 g of fresh soil was added to 90 mL of sterilized NaCl (0.85%) solution and shaken at 200 rpm min$^{-1}$ for 30 min. Ten-fold serial dilutions were prepared, and each well of the Biolog Eco-plates was inoculated with 150 µL of the $10^{-2}$ suspension. The plates were incubated at 30 °C for 10 days, and the color development was read as absorbance every 24 h with an automated plate reader (VMAX, Molecular Devices, Crawley, UK) at a wavelength of 590 nm. The 72 h absorbance values were used to calculate the average well color development (AWCD) and indicated the microbial metabolic activity.

Soil DNA was extracted from each sample using the PowerSoil kit (MoBioLaboratories, Carlsbad, CA, USA) according to the manufacturer's instructions. The quality of the purified DNA was assessed based on the 260/280 nm and 260/230 nm absorbance ratios obtained, using a NanoDrop ND-1000 spectrophotometer (NanoDrop Technologies Inc., Wilmington, DE, USA). The DNA was stored at −80 °C until use.

An aliquot of the extracted DNA from each sample was used as a template for amplification. The V3–V4 hypervariable regions of bacterial 16S rRNA genes were

amplified using the primers 338F 5′-barcode-ACTCCTACGGGAGGCAGCAG-3′ and 806R 5′-GGACTACHVGGGTWTCTAAT-3′. PCR reactions were performed in triplicate with a 20 µL mixture containing 4 µL of 5× FastPfu Buffer, 2 µL of 2.5 mM dNTPs, 0.8 µL of each primer (5 µM), 0.4 µL of FastPfuPolymerase, and 10 ng of template DNA. The following thermal program was used for amplification: 95 °C for 3 min, followed by 27 cycles at 95 °C for 30 s, 55 °C for 30 s, and 72 °C for 45 s and a final extension at 72 °C for 10 min. PCR amplicons were extracted from 2% agarose gels and purified using an AxyPrep DNA Gel Extraction Kit (Axygen Biosciences, Union City, CA, USA) according to the manufacturer's instructions and quantified using QuantiFluor$^{TM}$ -ST (Promega, USA). The purified amplicons from all samples were pooled at equimolar concentrations. Sequencing was conducted on an Illumina MiSeq platform at Majorbio Bio-Pharm Technology Co., Ltd. (Shanghai, China).

Raw sequences > 200 bp with an average quality score > 20 and without ambiguous base calls were quality processed, using the Quantitative Insights into Microbial Ecology (QIIME) pipeline (version 1.17). Operational taxonomic units (OTUs) were clustered with a 97% similarity cutoff using UPARSE (version 7.1 http://drive5.com/uparse/). The taxonomic assignment was performed using the Ribosomal Database Project (RDP) classifier (http://rdp.cme.msu.edu/). To correct for sampling effort (number of analyzed sequences per sample), we used a randomly selected subset of 19,460 sequences per sample for subsequent analysis.

## Root traits

The fine root samples (diameter <1 mm, including roots for absorption and transportation) were selected for scanning on a desktop scanner, and images were processed with WinRHIZO (Regent Instruments Inc., Quebec City, QC, Canada) to determine the average root diameter and total root length. These roots were then oven-dried to a constant weight. Specific root length (SRL) was calculated as the ratio of total root length to root dry weight, and root tissue density (RTD) was calculated as the ratio of root dry weight to root volume.

## Data analysis

For the analyses of bacterial community composition and function similarity, we calculated pairwise environmental distances (Euclidean distance) and a pairwise community Bray–Curtis dissimilarity matrix for the whole set of bacterial OTUs and Biolog data within the *vegan* package using R (R Core Team, 2016). Mantel tests (10,000 permutations) were used to explore the significance of the influence of geographical distance on Bray–Curtis dissimilarities.

We used structural equation modeling (SEM) to evaluate the direct and indirect relationships between geographical distance, soil properties, root traits, and bacterial community composition and functions. First, we established an a *priori* model based on the known effects and relationships among the drivers of community composition and function. Then we parameterized the model using our dataset and tested its overall goodness of fit. We used the $\chi^2$-test and root mean square error of approximation

(RMSEA). Furthermore, we calculated the standardized total effects of distance, soil properties and root traits on soil bacterial community composition and function. All the SEM analyses were conducted using AMOS 20.0 (AMOS IBM, USA) (*Grace & Keeley, 2006*).

# RESULTS

## Spatial variability of soil properties, root traits and bacterial community

We identified a total of 1,233,787 high-quality bacterial sequences grouped into 10,739 OTUs. The average number of bacterial sequences per sample was 23,279, which were classified as 2,311 to 3,402 OTUs (with an average of $2,901 \pm 32$ OTUs, Table S1). The bacterial alpha diversity (Shannon index) varied from 5.73–6.84, with an average of 6.28. The dominant phyla of bacterial communities across all soil samples were Proteobacteria, Acidobacteria, Actinobacteria, Chloroflexi, Verrucomicrobia, Bacteroidetes, Nitrospirae and Gemmatimonadetes (relative abundance >1%, Fig. 1), which accounted for more than 95% of the bacterial sequences. Alphaproteobacteria and Acidobacteria were most abundant at the class level, and the dominant classes (relative abundance>2%) also included Spartobacteria, Thermoleophilia, Actinobacteria, Deltaproteobacteria, Betaproteobacteria, and the other five classes, accounting for about 85% of the bacterial sequences (Fig. 1).

Metabolic activity (indicated as AWCD) and diversity varied from 0.51–1.57, and 2.19–3.35 with an average of 1.00 and 2.59, respectively (Table S1). Despite the fine scale of the research site (only $30 \times 40$ m$^2$), the root traits, soil parameters, bacterial community composition, and metabolic functions presented a high degree of spatial variance (Table S1). While the CVs of pH and Shannon–Wiener diversity index of bacterial community composition were relatively small, with CVs <20%, the other parameters had a high level of variance (>20%).

Microbial metabolic activities were related to the abundance of multiple bacterial classes in the phyla of Verrucomicrobia, Proteobacteria, Planctomysetes, Cyanobacteria, Chloroflexi, Bacteroidetes, and Actinobacteria (Fig. 2). The stepwise regression analysis showed that the six metabolic groups were all related to soil pH (28.9–57.8%), DOC (17.1–32.1%) and various bacterial classes (16.8–53.5%) (Tables S2, S3).

## Distance-decay patterns of bacterial community composition and metabolic functions

Dissimilarities (Bray Curtis index) in the bacterial community composition and metabolic functions were positively correlated with geographic distance (Mantel $r = 0.194$, $p < 0.05$; Mantel $r = 0.119$, $p < 0.05$) (Fig. 3). At the phylum level, besides Proteobacteria and Actinobacteria, the other dominant bacterial groups (Acidobacteria, Chloroflexi, Verrucomicrobia, Bacteroidetes, Nitrospirae, and Gemmatimonadetes) all showed distance-decay patterns (Fig. 4). Microbial metabolic activities towards carbohydrates, carboxylic acids, polymers, and amines presented distance-decay patterns (Fig. 5).

The multivariate Mantel correlogram showed that for bacterial community composition, the first three distance classes had a positive autocorrelation ($p < 0.05$; i.e., up to 16.1 m),
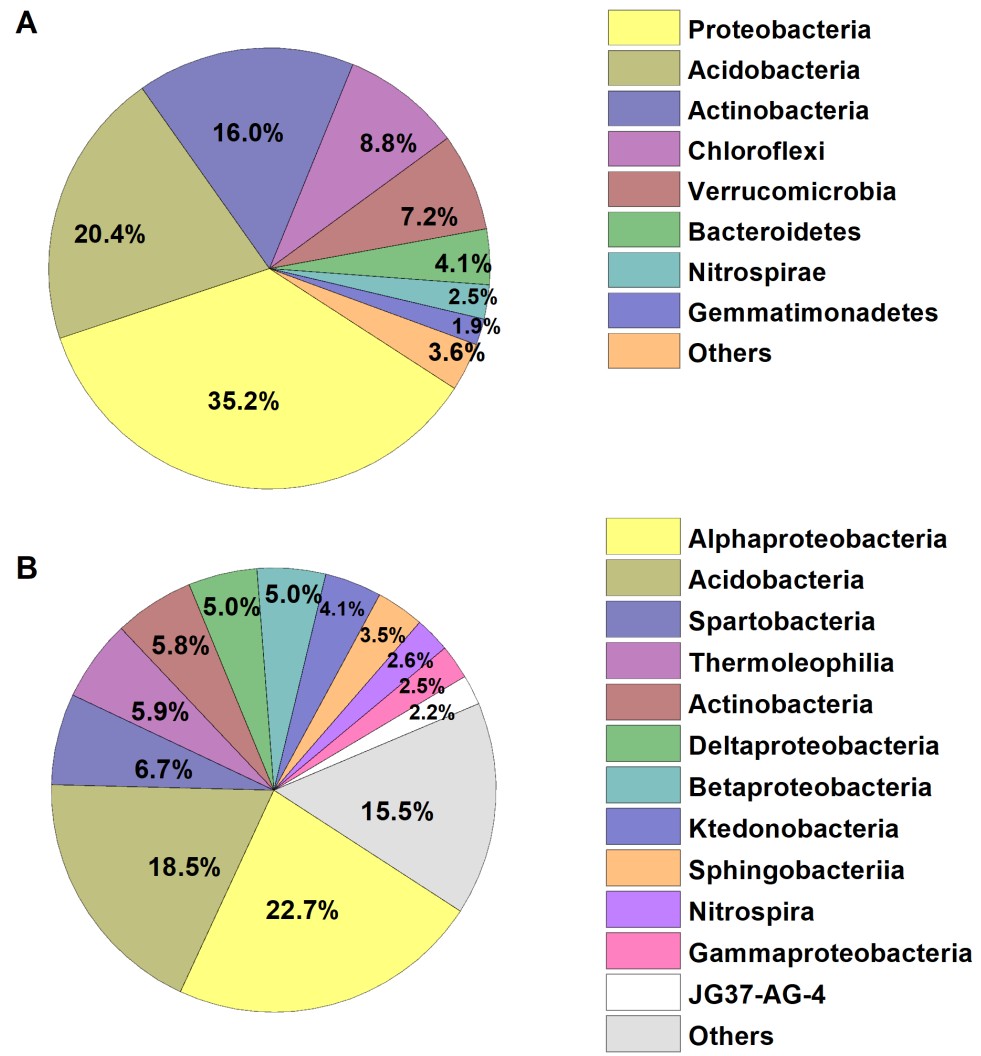

**Figure 1** Relative abundances of the dominant bacterial (A) phylum and (B) class in the broad-leaved Korean pine forest. The relative abundances are based on the proportional frequencies of the classified DNA sequences.

while the next two classes have a negative autocorrelation ($p < 0.05$; i.e., up to 26.5 m) (Fig. 6). No significant autocorrelations were found for the further distance class. However, the correlogram showed a sudden significant decrease in autocorrelation ($p < 0.05$) only at the smallest distance class (i.e., up to 5.7 m) (Fig. 6). This pattern indicated an abrupt change in metabolic functions with increasing distance.

## Drivers of bacterial community composition and carbon metabolic functions

Soil properties and root traits (root biomass and SRL) together shaped the bacterial community and affected metabolic activity (Figs. 7 and 8). The bacterial diversity (H′) was related to pH ($p < 0.01$) and C/N ($p < 0.05$), while the functional diversity was only
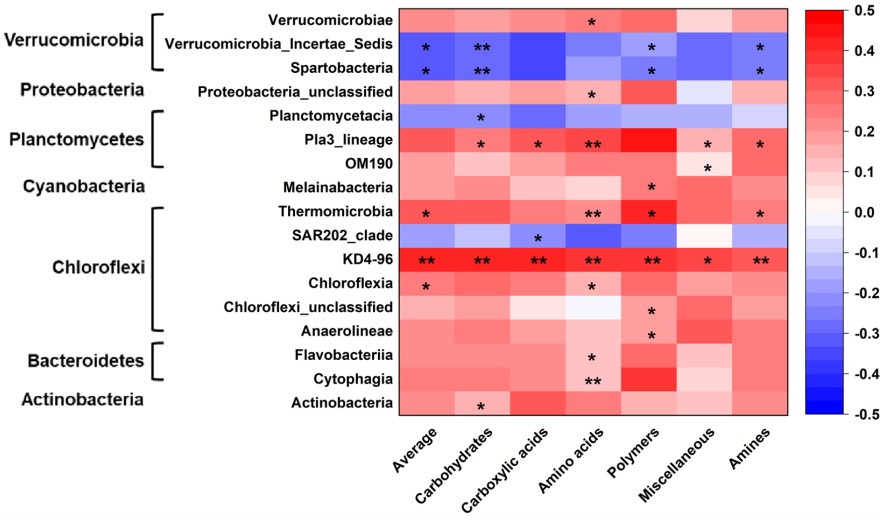

**Figure 2** Correlations of bacterial groups and metabolic functions. $*p \leq 0.05$; $**p \leq 0.01$. The blue-red bar on the right shows the negative-positive correlations.

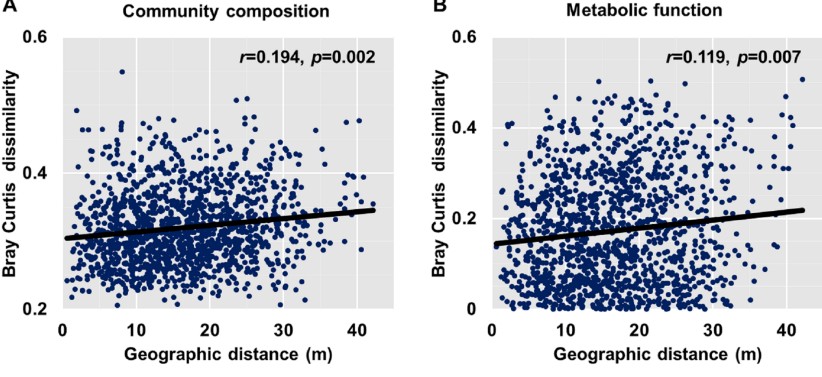

**Figure 3** Relationship between Bray–Curtis community dissimilarity and geographic distance for (A) bacterial community composition and (B) metabolic functions. Each data point represents the Bray–Curtis dissimilarity score for two samples and the geographic distance between the samples.

related to DOC concentration ($p < 0.01$) (Fig. 7). Bacterial community composition and functional groups were related to various soil and root properties (Fig. 7). The relative abundances of bacterial groups were mainly related to pH, SOC, TN, DOC, and SRL, while the metabolic groups were related to pH, SOC, TN, C/N, DOC, POC and PON (Fig. 7).

The SEMs explained 37.9% and 63.1% of the variance found in bacterial community composition and metabolic functions (Fig. 8), respectively. The distance only affected the bacterial community directly via its influence on soil properties (Fig. 8). Root biomass had a positive effect on bacterial community composition. SOC, C/N, pH, and root biomass collectively contributed to the variance of bacterial community composition, among which SOC and root biomass contributed the most (Fig. 8A). In contrast, pH was the

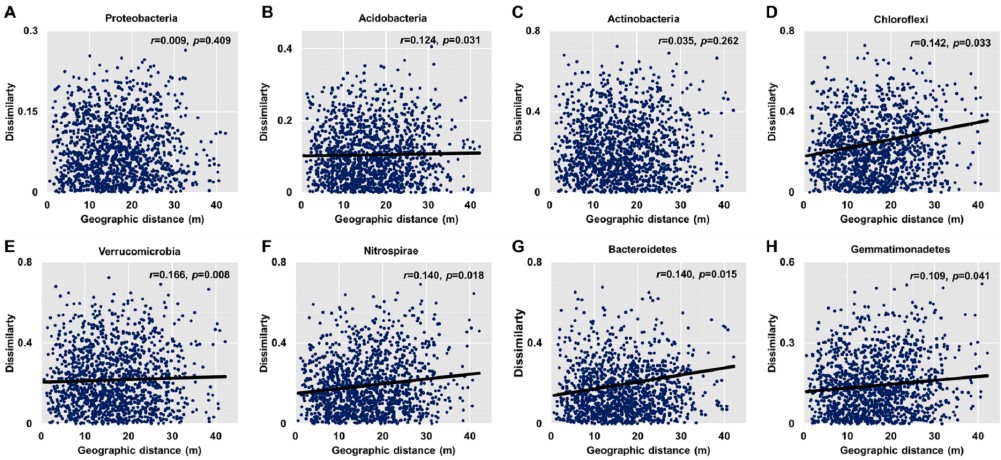

**Figure 4** **Relationships between dissimilarity and geographic distance for dominant bacterial phyla.**
(A) Proteobacteria. (B) Acidobacteria. (C) Actinobacteria. (D) Chloroflexi. (E) Verrucomicobia. (F) Nitrospirae. (G) Bacteroidetes. (H) Gemmatimonadetes. Each data point represents the Bray–Curtis dissimilarity score for two samples and the geographic distance between the samples.

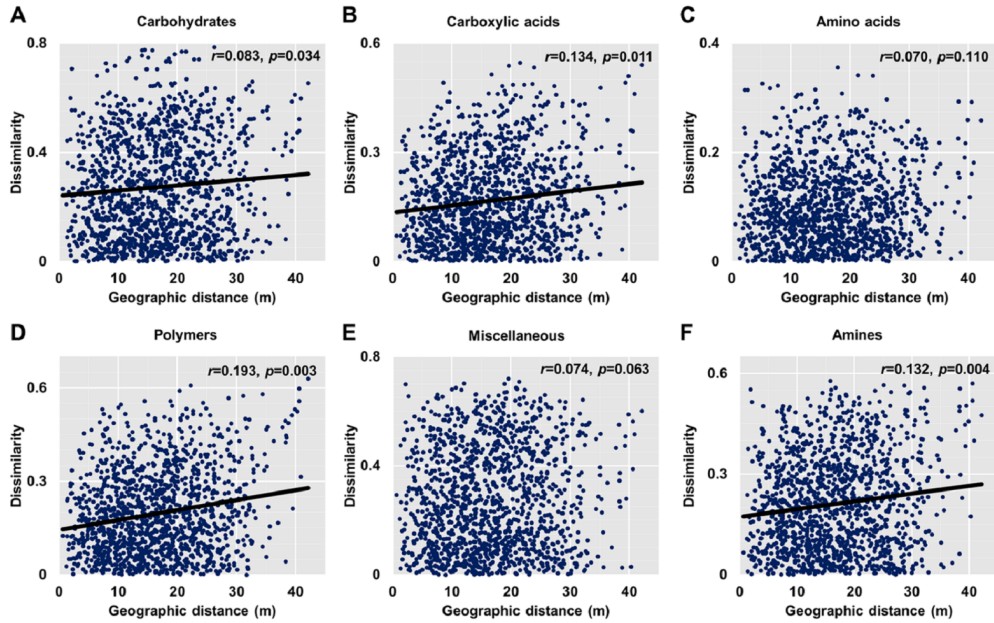

**Figure 5** **Relationships between dissimilarity and geographic distance for six metabolic functions.**
(A) Carbohydrates. (B) Carboxylic acids. (C) Amino acids. (D) Polymers. (E) Miscellaneous. (F) Amines.
Each data point represents the Bray–Curtis dissimilarity score for two samples and the geographic distance between the samples.

dominant driver in determining the variance of metabolic functions (Fig. 8B). Specifically, the root traits made a difference in bacterial community composition but were irrelevant to functional differences.

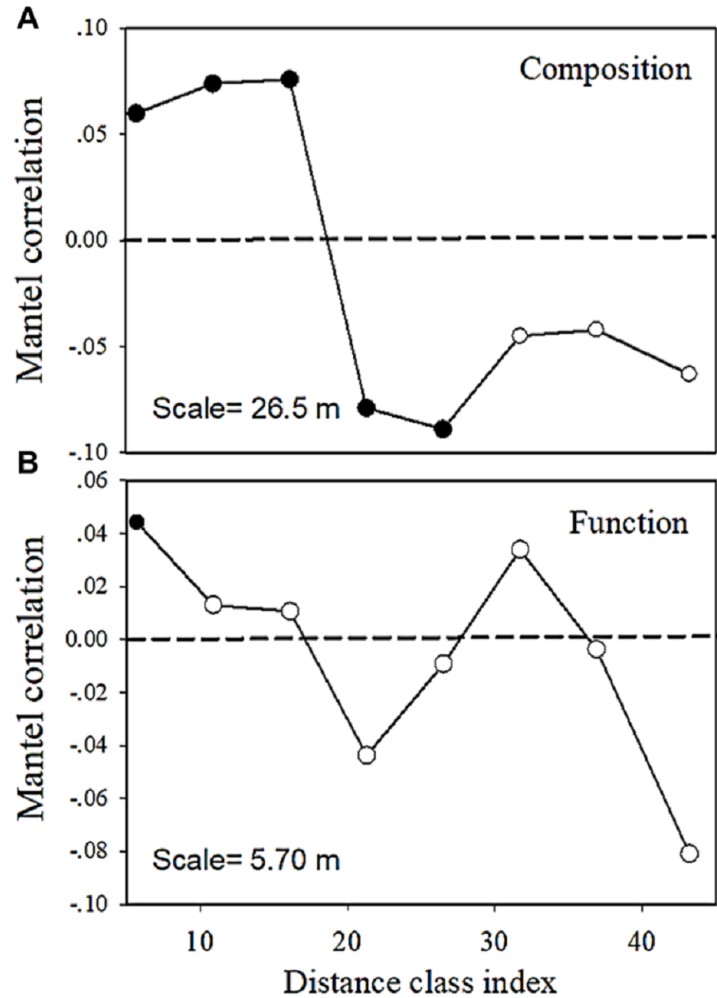

**Figure 6  Multivariate Mantel correlograms showing the significance of spatial autocorrelation in (A) bacterial composition and (B) functions.** Solid black points represent scales with significant ($p < 0.05$) spatial autocorrelation (positive Mantel correlation values) or spatial clustering (negative Mantel correlation values). Open points represent non-significant values. Holm's correction was applied for multiple comparisons. Therefore, 'scale values' on the plots provide the approximate distances on the correlograms, where spatial autocorrelation in bacterial composition or functions between samples becomes non-significant (that is, only communities separated by distances greater than the scale values are likely differ significantly).

## DISCUSSION

Soil bacteria are the most abundant and diverse group of organisms on Earth, driving many ecosystem processes (*Bardgett & Wh, 2014*; *Delgado-Baquerizo et al., 2018b*; *Sugden, 2018*). Understanding bacterial biogeographical patterns and drivers is crucial for resolving the complex and coordinated microbial mechanisms of maintaining soil nutrient cycling. Distance-decay relationships exist in the distribution of bacterial communities (*Fierer &*

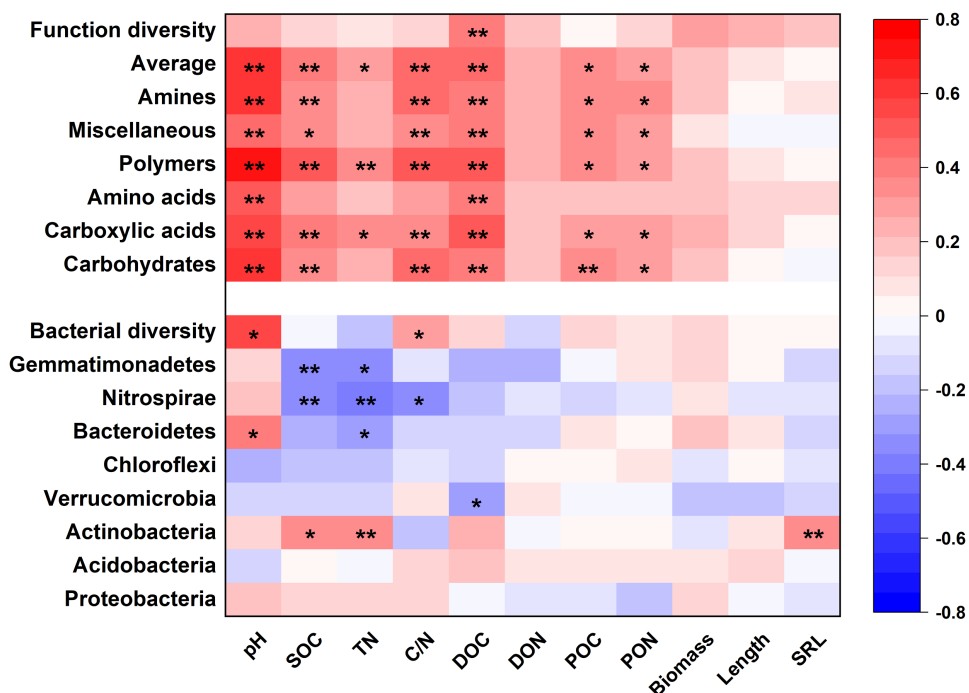

**Figure 7** The heatmap of the Pearson's correlation coefficients of the relative abundances of dominant bacterial groups and metabolic functions with plant and soil properties. *$p \leq 0.05$; **$p \leq 0.01$.

*Jackson, 2006*; *Ranjard et al., 2013*; *Sugden, 2018*). Nevertheless, few investigations have yet been conducted at fine spatial scales.

A correlogram that declines from significant positive correlations at short distances to significant negative correlations at large distances is consistent with a patchy spatial distribution (*Legendre & Fortin, 1989*). We confirmed the distance-decay patterns of bacterial communities and functions at smaller spatial scales than most previous studies of forest soil. These findings improved our understanding of bacterial community variability at a fine scale and provided appropriate scales to monitor the microbial communities of forest soil. Furthermore, these results showed that the bacterial community composition and metabolic functions presented an obvious asymmetric variance (Fig. 6), which could be due to the dispersal in the bacterial community was higher than that of microbes responsible for metabolic functions (*Lear et al., 2014*). Certain microbial community functions are in fact driven more by low levels of the bacterial groups with relatively limited dispersal ability (*Lear et al., 2014*; *Severin, Ostman & Lindstrom, 2013*), while other abundant groups that are less responsible for community functions had more ability to disperse and colonize new habitats (*Lear et al., 2014*).

The spatial heterogeneity mainly arose from the soil properties and root traits, rather than distance or dispersal limitations at the fine scale. The spatial patterns in the soil bacterial community at the fine scale were mainly due to soil properties and root traits (Fig. 8). The parameters we measured explained 37.9% and 63.1% of the variance of soil bacterial community composition and metabolic functions, respectively. Our findings are in line with
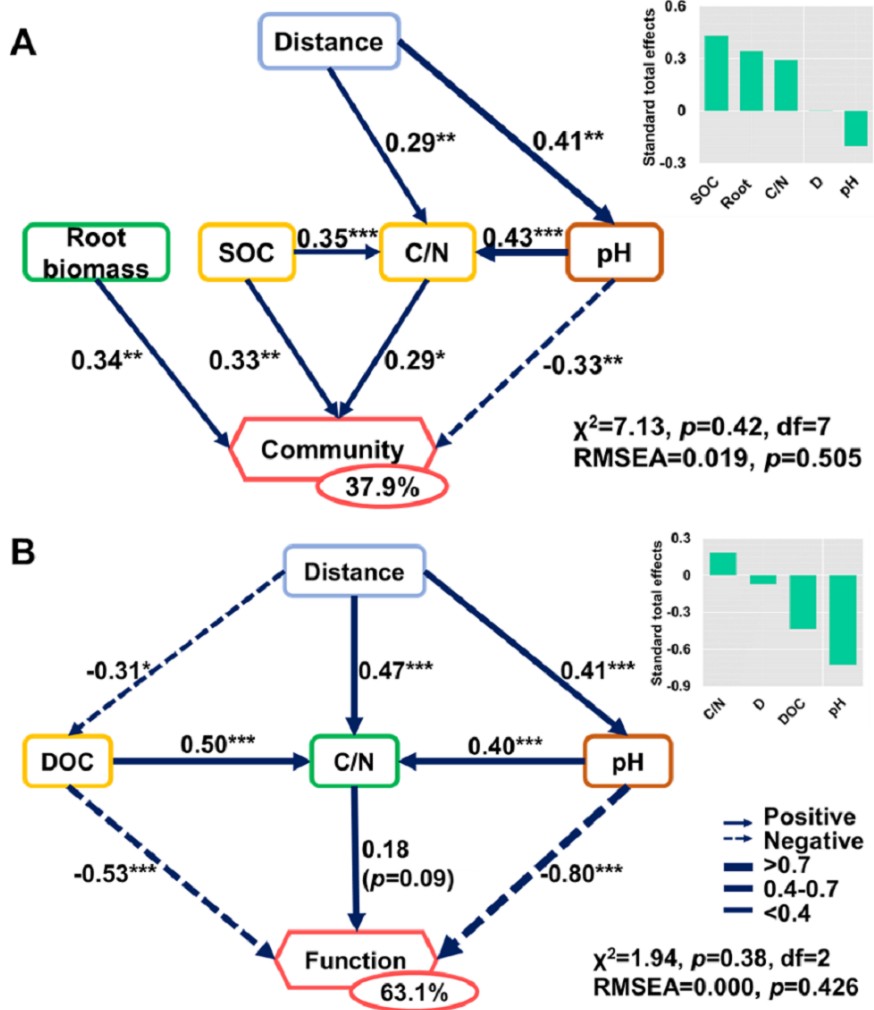

**Figure 8   Direct and indirect effects of soil nutrients and root traits on beta-diversity of bacterial community composition and metabolic functions.** Structural equation models are shown for the (A) bacterial community and (B) metabolic functions. Arrows represent causal relationships. Numbers on arrows are standardized path coefficients. Percentages in circles indicate the variance explained by the model ($R^2$). Asterisks denote the level of significance: $*p \leq 0.05$; $**p \leq 0.01$; $***p \leq 0.001$.

previous research demonstrating the dominant effects of pH in shaping the soil bacterial community of Changbai Mountain (*Shen et al., 2013*). The importance of pH in shaping soil bacterial communities has been studied at various scales (*Fierer & Jackson, 2006*; *Liu et al., 2014*; *Tripathi et al., 2018*) and all results indicated the pivotal role of pH in controlling bacterial communities (*Delgado-Baquerizo et al., 2018b*; *Rousk, Brookes & Bååth, 2010*; *Shen et al., 2013*). Some explanations may explain the effects of soil pH on metabolic functions. First, soil pH impacts the substrates and microenvironment for metabolic reactions (*Berg & Mcclaugherty, 2013*; *Jones et al., 2019*), which change the microbial metabolic activity directly. Second, soil pH changes the abundance and activity of microbes participating in the metabolic reaction (*Berg & Mcclaugherty, 2013*; *Jones et al., 2019*).

The effects of soil pH on metabolic functions are the result of the combined action of various factors. In addition to pH, the effects of other soil properties, such as soil organic matter content are important (*Delgado-Baquerizo et al., 2018b*; *Lladó, López-Mondéjar & Baldrian, 2017*; *Tian et al., 2015*).

The plant communities influence the below-ground communities by litterfall and root rhizodeposition. Fine roots and their symbiotic microorganisms play important roles in soil nutrient availability and soil organic matter decomposition (*Finzi et al., 2015*; *Han et al., 2020*; *López-Angulo et al., 2020*; *Saleem et al., 2020*). Root traits reflect the quantity and quality of root litter and exudate transferred into the soil organic matter pool (*Henneron et al., 2020*; *Klimešová, Martínková & Ottaviani, 2018*; *See et al., 2019*) and decomposed by soil microbes. However, we only found that the root biomass contributed to variance in bacterial community composition but were uncorrelated with carbon metabolic functions, and there was no significant relationship between root traits and SOC. This may be due to the dual effects of roots on SOC (*Dijkstra, Zhu & Cheng, 2020*), in which roots increase SOC in forms of root litter and exudate, but also carbon input from roots promotes SOC decomposition because of the priming effect. Thus, we did not find a significant relationship between root biomass and SOC, but both affected soil the bacterial community directly. Larger root biomass per unit area implied more rhizodeposition that promoted certain groups of bacterial growth, such as the Actinobacteria phylum, which resulted in variance of bacterial community composition. On the other hand, metabolic functions were driven more by low levels of the bacterial groups, and those root-affected groups were less responsible for metabolic functions, leading to different effects of root traits on community composition and functions.

## CONCLUSIONS

The bacterial metabolic functions and community composition varied significantly at a scale of a few meters and tens of meters, respectively, due to the heterogeneity of forest soil. Soil nutrient contents (SOC and C/N), pH, and root biomass together accounted for 37.9% of the variance in bacterial community composition, while only pH and nutrients contributed to 63.1% of the variance in metabolic functions. Root traits only affected community composition, but made no difference in the variance of metabolic functions. Geographical distance had only indirect effects via soil properties. This finding revealed that the synthesis of soil-roots-microbes should be think comprehensively in future studies.

### Funding

This work was supported by the National Natural Science Foundation of China (Grant No. 31770560), and the Major Program of the National Natural Science Foundation of China (Grant No. 2017YFA0604803). The funders had no role in study design, data collection and analysis, decision to publish, or preparation of the manuscript.

## Grant Disclosures

The following grant information was disclosed by the authors:
National Natural Science Foundation of China: 31770560.
Major Program of the National Natural Science Foundation of China: 2017YFA0604803.

## Competing Interests

The authors declare there are no competing interests.

## Author Contributions

- Jialing Teng and Jing Tian conceived and designed the experiments, performed the experiments, analyzed the data, prepared figures and/or tables, authored or reviewed drafts of the paper, and approved the final draft.
- Guirui Yu and Yakov Kuzyakov conceived and designed the experiments, authored or reviewed drafts of the paper, and approved the final draft.

## Data Availability

Data is available at GenBank: PRJNA663422.

The sequences are also available at Figshare:

Teng, Jialing (2021): ROWDATA.zip. figshare. Dataset. https://doi.org/10.6084/m9.figshare.12979112.v1.

## Supplemental Information

Supplemental information for this article can be found online at http://dx.doi.org/10.7717/peerj.10902#supplemental-information.

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
