# Peer review of "Soil properties and root traits jointly shape fine-scale spatial patterns of bacterial community and metabolic functions within a Korean pine forest"

_PeerJ, doi:10.7717/peerj.10902_

## Round 0.1 · original submission · Major Revisions

I have received reports from our advisors on your manuscript and they have suggested major revisions before your manuscript is further considered for publication in PeerJ. I have accepted their advice and invite you to revise your manuscript. If you are ready to undertake the revisions, please give careful consideration to the comments and suggestions from reviewers while preparing the revised version of the manuscript. I look forward to receiving the revised version of your manuscript.

·

Basic reporting

See below section.

Experimental design

See below section.

Validity of the findings

See below section.

Additional comments

Dear Author:
The manuscript “Soil properties and root traits jointly shape fine-scale spatial patterns of bacterial community and metabolic functions within a Korean pine forest” is quite interesting and informative. The methods and experimental design seem reasonable and sound. However, there are few concerns with this manuscript which need proper consideration. Such as:

1. Why you took samples from 0-10cm ? usually the surface sampling is done from 0-15cm.
2. Line 122: Based on which classification system the soil is classified as Haplic Andosols?
3. Line 136: How the temperature in the color box was kept 4 °C?
4. How many samples were collected per plot (30 m × 40 m)?
5. Line 142: Rephrase the sentence “Those for soil organic matter and particulate organic matter analyses were air dried”.
6. Line 145: The air dried sample passed…. should be replaced by “ the air dried samples were passed through 2 mm sieve” .
7. Line 145: The word “measured for” should be replaced by “analyzed for”.
8. Line 144-154; never start the sentence with abbreviation.
9. Line 202: what temperature was used for oven drying of roots?
10. Why pH and nutrient contributed to 63% of variance of metabolic functions ? Add the pH and nutrients data for all samples as a table which can explain such a huge variation?
11. What could be the possible reasons for the largest variance in metabolic functions due to soil pH? Explain in discussion section.
12. What are the possible reasons for such a significant difference in soil microbial composition even at a small distance?
13. The literature used is up to 2016. Try to add latest references in discussion section.

Reviewer 2 ·

Basic reporting

The manuscript reports the spatial shift in the bacterial community in response to the soil properties and root traits. The authors reported after rigorous analysis of Illumina 16S rRNA amplicon sequencing that soil organic matter and roots biomass are the main drivers of bacterial community composition in tested samples. Furthermore, after the analysis of metabolic functions through Biolog plates, it was found that pH is the main factor responsible for the variation of metabolic properties of bacterial communities. Overall, the manuscript seems to be well-composed and may be accepted after minor revisions.

Experimental design

The experimental design is fine.

Validity of the findings

Good

Additional comments

1) In the introduction, authors should specify the introduction and background to bacterial communities and not microbial communities since they tested only bacterial communities
2) The metabolic variation (functional variation) on the basis of Biolog Eco plates was measured with soil samples probably having a complex mixture of microorganisms; however, the genetic variation was based only on the 16S rRNA gene representing the bacterial portion of complex communities, how authors could correlate the findings of both functional and genetic variation?
3) Authors should specify the term microbial community in case of metabolic measurements (Biolog plates) and bacterial communities in case of genetic variation at a fine-scale level
4) The discussion is supported by old references at some places, these references may be replaced with updated literature

Reviewer 3 ·

Basic reporting

Teng et al manuscript reports interesting findings about the role of root traits and soil properties in structuring soil microbes across different geographic distances. Overall, manuscripts reports very convincing and interesting results, and below are my suggestions that may improve the current work.
Major: Novelty: many studies have reported the role of soil pH in microbial communication? How you results are different from previous studies?
Minor
Ln 33. “environmental variables” which environmental variable you thing is prominent in determining microbial communities?
At some level manuscript is too wordy and sentences are full of redundant phrases (e.g. ln 43-44, first portion of ln 47). Authors may want to check and address this issues.
Ln 58/ln94. For very descriptive and established facts, authors have unnecessarily cited bulk of references, which are even longer than the contents of actual sentences, authors may want to check n correct citation.
Ln 81. I appreciate that the reviewers have acknowledged that “Much less is known, however, about the role of roots in regulating the microbial communities”. BUT again, the cited references have little to nothing to do with the contents of this particular notion. Rather authors may want to support their root traits aspects with relevant literature (e.g. Rhizosphere 16 (2020): 100249).
Ln 106. You may want to clarify which “metabolic functions” are you going to study?
Ln 227. What does “with an average of 6.28” this mean? What is the unit?

Ln 275. Did you test, what was the relationship between SOC and root biomass? Some recent studies showed a positive relationship between SOC and root biomass(Rhizosphere 16 (2020): 100248.). Please discuss this aspect, while linking root traits with microbial communities.

Discussion: you may want to link root traits with microbial communities using relevant literature. You discussion is full of general sentences, which are loaded with references(not necessarily relavent).

Figures are blurred and their quality is not good.

Overall, it is a very well-performed study.

Experimental design

Teng et al manuscript reports interesting findings about the role of root traits and soil properties in structuring soil microbes across different geographic distances. Overall, manuscripts reports very convincing and interesting results, and below are my suggestions that may improve the current work.
Major: Novelty: many studies have reported the role of soil pH in microbial communication? How you results are different from previous studies?
Minor
Ln 33. “environmental variables” which environmental variable you thing is prominent in determining microbial communities?
At some level manuscript is too wordy and sentences are full of redundant phrases (e.g. ln 43-44, first portion of ln 47). Authors may want to check and address this issues.
Ln 58/ln94. For very descriptive and established facts, authors have unnecessarily cited bulk of references, which are even longer than the contents of actual sentences, authors may want to check n correct citation.
Ln 81. I appreciate that the reviewers have acknowledged that “Much less is known, however, about the role of roots in regulating the microbial communities”. BUT again, the cited references have little to nothing to do with the contents of this particular notion. Rather authors may want to support their root traits aspects with relevant literature (e.g. Rhizosphere 16 (2020): 100249).
Ln 106. You may want to clarify which “metabolic functions” are you going to study?
Ln 227. What does “with an average of 6.28” this mean? What is the unit?

Ln 275. Did you test, what was the relationship between SOC and root biomass? Some recent studies showed a positive relationship between SOC and root biomass(Rhizosphere 16 (2020): 100248.). Please discuss this aspect, while linking root traits with microbial communities.

Discussion: you may want to link root traits with microbial communities using relevant literature. You discussion is full of general sentences, which are loaded with references(not necessarily relavent).

Figures are blurred and their quality is not good.

Overall, it is a very well-performed study.

Validity of the findings

Teng et al manuscript reports interesting findings about the role of root traits and soil properties in structuring soil microbes across different geographic distances. Overall, manuscripts reports very convincing and interesting results, and below are my suggestions that may improve the current work.
Major: Novelty: many studies have reported the role of soil pH in microbial communication? How you results are different from previous studies?
Minor
Ln 33. “environmental variables” which environmental variable you thing is prominent in determining microbial communities?
At some level manuscript is too wordy and sentences are full of redundant phrases (e.g. ln 43-44, first portion of ln 47). Authors may want to check and address this issues.
Ln 58/ln94. For very descriptive and established facts, authors have unnecessarily cited bulk of references, which are even longer than the contents of actual sentences, authors may want to check n correct citation.
Ln 81. I appreciate that the reviewers have acknowledged that “Much less is known, however, about the role of roots in regulating the microbial communities”. BUT again, the cited references have little to nothing to do with the contents of this particular notion. Rather authors may want to support their root traits aspects with relevant literature (e.g. Rhizosphere 16 (2020): 100249).
Ln 106. You may want to clarify which “metabolic functions” are you going to study?
Ln 227. What does “with an average of 6.28” this mean? What is the unit?

Ln 275. Did you test, what was the relationship between SOC and root biomass? Some recent studies showed a positive relationship between SOC and root biomass(Rhizosphere 16 (2020): 100248.). Please discuss this aspect, while linking root traits with microbial communities.

Discussion: you may want to link root traits with microbial communities using relevant literature. You discussion is full of general sentences, which are loaded with references(not necessarily relavent).

Figures are blurred and their quality is not good.

Overall, it is a very well-performed study.

Additional comments

Teng et al manuscript reports interesting findings about the role of root traits and soil properties in structuring soil microbes across different geographic distances. Overall, manuscripts reports very convincing and interesting results, and below are my suggestions that may improve the current work.
Major: Novelty: many studies have reported the role of soil pH in microbial communication? How you results are different from previous studies?
Minor
Ln 33. “environmental variables” which environmental variable you thing is prominent in determining microbial communities?
At some level manuscript is too wordy and sentences are full of redundant phrases (e.g. ln 43-44, first portion of ln 47). Authors may want to check and address this issues.
Ln 58/ln94. For very descriptive and established facts, authors have unnecessarily cited bulk of references, which are even longer than the contents of actual sentences, authors may want to check n correct citation.
Ln 81. I appreciate that the reviewers have acknowledged that “Much less is known, however, about the role of roots in regulating the microbial communities”. BUT again, the cited references have little to nothing to do with the contents of this particular notion. Rather authors may want to support their root traits aspects with relevant literature (e.g. Rhizosphere 16 (2020): 100249).
Ln 106. You may want to clarify which “metabolic functions” are you going to study?
Ln 227. What does “with an average of 6.28” this mean? What is the unit?

Ln 275. Did you test, what was the relationship between SOC and root biomass? Some recent studies showed a positive relationship between SOC and root biomass(Rhizosphere 16 (2020): 100248.). Please discuss this aspect, while linking root traits with microbial communities.

Discussion: you may want to link root traits with microbial communities using relevant literature. You discussion is full of general sentences, which are loaded with references(not necessarily relavent).

Figures are blurred and their quality is not good.

Overall, it is a very well-performed study.

---

## Round 0.2 · Minor Revisions

Thank you for revising the manuscript in accordance with the suggestions from our advisors. The scientific content of your manuscript is acceptable but the English still requires some improvement. Please consider using a professional editing service.

·

Basic reporting

I have already reviewed the manuscript “Soil properties and root traits jointly shape fine-scale spatial patterns of bacterial community and metabolic functions within a Korean pine forest” on October 19, 2020 and my decision was major revision. I am again of the opinion that the manuscript is well written and explores some new knowledge. I am impressed an in favor of publishing it in your esteem journal. However, there are some suggestions/ comments/ changes which must be adopted before its final publication.

Experimental design

The design of the study is well appropriate and there is no need for its further improvement.

Validity of the findings

The findings are valid and well supported by literature.

Additional comments

I have already reviewed the manuscript “Soil properties and root traits jointly shape fine-scale spatial patterns of bacterial community and metabolic functions within a Korean pine forest” on dated October 19, 2020 and my decision was major revision. I am again of the opinion that the manuscript is well written and explores some new knowledge. I am impressed an in favor of publishing it in your esteem journal. However, there are some suggestions/ comments/ changes which must be adopted before its final publication. My comments are appended below:

• The abstract has some repetitions which needs to be avoided like “Root biomass explained the second largest variance in soil bacterial community composition after SOC”
• Introduction is well written and supports the study but there is a need of English language editing as well as new literature.
• Why soil depth was kept 0-10cm? Mostly we take surface soil up to 15cm.

My detail comments have already been submitted to you October 19, 2020. In addition to the above comments, those submitted earlier must be properly adopted before making final decision.

Reviewer 2 ·

Basic reporting

The manuscript reports the spatial shift in the bacterial community in response to the soil properties and root traits. The authors reported after rigorous analysis of Illumina 16S rRNA amplicon sequencing that soil organic matter and roots biomass are the main drivers of bacterial community composition in tested samples. Furthermore, after the analysis of metabolic functions through Biolog plates, it was found that pH is the main factor responsible for the variation of metabolic properties of bacterial communities. Overall, the manuscript seems to be well-composed and may be accepted after minor revisions.

Experimental design

The experimental design is fine

Validity of the findings

Good

Additional comments

The authors have addressed my comments and the MS is now ready for publication

Reviewer 3 ·

Basic reporting

Authors have sufficiently addressed the raised concerns and manuscript can be considered for publication.

Experimental design

Authors have sufficiently addressed the raised concerns and manuscript can be considered for publication.

Validity of the findings

Authors have sufficiently addressed the raised concerns and manuscript can be considered for publication.

Additional comments

Authors have sufficiently addressed the raised concerns and manuscript can be considered for publication.

---

## Round 0.3 · accepted · Accept

Thank you for revising and improving the manuscript. I have now accepted the manuscript for publication.